# Medermycin Inhibits TNFα-Promoted Inflammatory Reaction in Human Synovial Fibroblasts

**DOI:** 10.3390/ijms241813871

**Published:** 2023-09-08

**Authors:** Sho Inoue, Yuki Inahashi, Makoto Itakura, Gen Inoue, Kyoko Muneshige, Tomoyasu Hirose, Masato Iwatsuki, Masashi Takaso, Masayuki Miyagi, Kentaro Uchida

**Affiliations:** 1Department of Orthopedic Surgery, Kitasato University School of Medicine, 1-15-1 Minami-ku Kitasato, Sagamihara City 252-0374, Japan; ino2510@gmail.com (S.I.); ginoue@kitasato-u.ac.jp (G.I.); shige_no27@yahoo.co.jp (K.M.); mtakaso@kitasato-u.ac.jp (M.T.); masayuki008@aol.com (M.M.); 2Graduate School of Infection Control Sciences, Kitasato University, 5-9-1 Minato-ku Shirokane, Tokyo 108-8641, Japan; y-ina@lisci.kitasato-u.ac.jp (Y.I.); thirose@lisci.kitasato-u.ac.jp (T.H.); iwatuki@lisci.kitasato-u.ac.jp (M.I.); 3Ōmura Satoshi Memorial Institute, Kitasato University, 5-9-1 Minato-ku Shirokane, Tokyo 108-8641, Japan; 4Department of Biochemistry, Kitasato University School of Medicine, 1-15-1 Minami-ku, Kitasato, Sagamihara City 252-0374, Japan; mitakura@med.kitasato-u.ac.jp; 5Shonan University of Medical Sciences Research Institute, Nishikubo 500, Chigasaki 253-0083, Japan

**Keywords:** medermycin, synovial fibroblast, NFκB, TNF-α, osteoarthritis

## Abstract

Synovial inflammation plays a crucial role in the destruction of joints and the experience of pain in osteoarthritis (OA). Emerging evidence suggests that certain antibiotic agents and their derivatives possess anti-inflammatory properties. Medermycin (MED) has been identified as a potent antibiotic, specifically active against Gram-positive bacteria. In this study, we aimed to investigate the impact of MED on TNFα-induced inflammatory reactions in a synovial cell line, SW-982, as well as primary human synovial fibroblasts (HSF) using RNA sequencing, rtRT-PCR, ELISA, and western blotting. Through the analysis of differentially expressed genes (DEGs), we identified a total of 1478 significantly upregulated genes in SW-982 cells stimulated with TNFα compared to the vehicle control. Among these upregulated genes, MED treatment led to a reduction in 1167 genes, including those encoding proinflammatory cytokines such as *IL1B*, *IL6*, and *IL8*. Pathway analysis revealed the enrichment of DEGs in the TNF and NFκB signaling pathway, further supporting the involvement of MED in modulating inflammatory responses. Subsequent experiments demonstrated that MED inhibited the expression of IL6 and IL8 at both the mRNA and protein levels in both SW982 cells and HSF. Additionally, MED treatment resulted in a reduction in p65 phosphorylation in both cell types, indicating its inhibitory effect on NFκB activation. Interestingly, MED also inhibited Akt phosphorylation in SW982 cells, but not in HSF. Overall, our findings suggest that MED suppresses TNFα-mediated inflammatory cytokine production and p65 phosphorylation. These results highlight the potential therapeutic value of MED in managing inflammatory conditions in OA. Further investigations utilizing articular chondrocytes and animal models of OA may provide valuable insights into the therapeutic potential of MED for this disease.

## 1. Introduction

Osteoarthritis (OA), a common degenerative disease, is a primary cause of disability [1]. This condition involves alterations in the articular cartilage, synovium, and subchondral bone and is characterized by pain, joint dysfunction, and loss of tissue integrity. Clinically, administration of analgesics and non-steroidal anti-inflammatory drugs (NSAIDs), symptomatic slow acting drugs of OA (SYSADOA) including chondroitin sulfate and glucosamine, surgical intervention using medical devices, and physical therapy are used as therapeutic approaches against OA [2,3]. To date, however, no evidence has appeared to indicate that nonsurgical treatment for osteoarthritis, including NSAIDs and physical therapy, is successful in reversing the course of the disease. Therefore, more specific therapeutic approaches need to be researched and developed.

In osteoarthritis, synovial inflammation plays a critical role in joint damage and pain [4,5]. The increase in proinflammatory cytokines such as IL1β, IL6, and TNFα during synovial inflammation accelerates the destruction of articular cartilage via matrix degradation enzymes [5,6]. The inducible transcription factor NFκB has a key function under inflammatory conditions. Notably, NFκB expression is elevated in the synovium if it affected by radiographic OA [7]. Proinflammatory cytokines, including IL6 and IL8, produced by the synovium in osteoarthritis are well-established triggers for the activation of catabolic genes, a process which occurs through mechanisms involving NFκB activation [8,9]. Moreover, these cytokines contribute to the experience of pain in OA [10,11]. Additionally, NFκB regulates the chemokine ligands that facilitate the recruitment of immune cells during inflammation in the synovium [12,13]. These various effects point to the development of NFκB inhibitors as a promising therapeutic approach for the treatment of OA.

Accumulated evidence suggests that some antibiotic agents and their derivatives have anti-inflammatory properties [14,15,16,17]. For example, the nematocidal antibiotic jietacin and its derivative exert anti-inflammatory properties via the suppression of NFκB in synovial cells [14]. With regard to medermycin (MED), however, a potent antibiotic isolated from Streptomyces sp. K73 with antibiotic activity against Gram-positive bacteria [18], the possibility that this agent has an anti-inflammatory effect has not been examined.

The aim of this study was to determine if MED can reduce inflammatory reactions and reveal the mechanism of action of its effect on human synovial fibroblasts.

## 2. Results

MED isolated from Streptomyces tanashiensis 144 MED structure is shown in Figure 1.

To assess the impact of MED on TNFα-stimulated proinflammatory production, we conducted RNA sequencing analysis using the synovial cell line SW-982. Differential expression genes (DEGs) analysis revealed that 1478 genes were significantly upregulated between the vehicle and TNFα-stimulated SW-982 (Figure 2A). Of these upregulated genes, 1167 genes—including those for proinflammatory cytokines (*IL1B*, *IL6*, *IL8*) and *AKT3*—were reduced by MED treatment (Appendix A). Pathway analysis showed that DEGs were enriched in the NFκB (KEGG ID, 04060) and TNF signaling pathways (KEGG ID, 04668) (Figure 2B and Appendix A).

To confirm the findings from the RNA sequencing analysis and investigate the impact of different MED concentrations, we performed rtRT-PCR and ELISA assays (Figure 3A–D). Treatment with hrTNFα significantly increased the expression of *IL1B* mRNA and the level of IL1β protein in the supernatant of SW-982 cells (*IL1B*, *p* < 0.001; IL1β, *p* = 0.013; Figure 3A,B). However, exposure to MED at concentrations of 62.5, 125, and 250 ng/mL inhibited this increase (62.5 ng/mL, *IL1B*, *p* = 0.043. IL1β, *p* = 0.006; 125 ng/mL, *IL1B*, *p* = 0.005, IL1β, *p* < 0.001; 250 ng/mL, *IL1B*, *p* = 0.002, IL1β, *p* < 0.001; Figure 3A,B).

Furthermore, stimulation of SW-982 cells with hrTNFα significantly increased the expression of *IL6* mRNA and the level of *IL6* protein in the supernatant (*IL6*, *p* < 0.001; IL6, *p* = 0.030; Figure 3C,D). However, exposure to MED at concentrations of 62.5, 125, and 250 ng/mL inhibited this increase (125 ng/mL, *IL6*, *p* < 0.001. IL6, *p* < 0.001; 250 ng/mL, *IL6*, *p* < 0.001, IL6, *p* < 0.001; Figure 3C,D).

Additionally, treatment with hrTNFα also led to significant increases in *IL8* mRNA expression and *IL8* protein levels in the supernatant (*IL8*, *p* = 0.002; IL8, *p* = 0.013; Figure 3E,F). Furthermore, exposure to 250 ng/mL MED effectively inhibited the increase in *IL8* expression (*IL8*, *p* = 0.013. *IL8*, *p* < 0.001. Figure 3E,F).

Based on these findings in SW-982 cells, we further investigated the impact of MED on human primary synovial fibroblasts (HSF) (Figure 4A–F). Treatment with hrTNFα significantly increased the expression of *IL1B* (*p* < 0.001), but this effect was significantly reduced by 125 and 250 ng/mL MED treatment (125 ng/mL, *IL1B*, *p* = 0.002; 250 ng/mL, *IL1B*, *p* < 0.001) in HSF (Figure 4A,C,F). However, levels of IL1β concentration in the samples were found to be below the detection limit of the ELISA, which is set at <2 pg/mL (Figure 4B), which is consistent with a previous study [14]. The expression of *IL6* and *IL8* mRNA, as well as the production of IL6 and IL8, increased following hrTNFα treatment (*IL6*, *p* < 0.001; *IL6*, *p* = 0.025; *IL8*, *p* < 0.001; *IL8*, *p* = 0.016; Figure 4D,F). Treatment with 250 ng/mL medermycin reduced the mRNA and protein levels of *IL6* and *IL8* (*IL6*, *p* = 0.004; IL6, *p* = 0.025; *IL8*, *p* = 0.014; IL8, *p* < 0.002; Figure 4E,F).

Pathway analysis suggested that MED regulates NFκB signaling in synovial cells. In contrast, previous studies reported that MED inhibited Akt signaling in a tumor cell line [19,20]. As Akt signaling could crosstalk with NF-kb signaling [21,22,23], we subsequently examined the effect of MED on the NFκB and Akt pathways. Expression of p65 did not differ between the hrTNFα and vehicle groups. However, reduced p65 expression was observed in the presence of 125 (*p* = 0.026) and 250 (*p* = 0.049) ng/mL MED compared to the hrTNFα groups. hrTNFα induced the phosphorylation of p65 (*p* < 0.001), which was suppressed in the presence of 250 ng/mL MED (*p* = 0.040) in SW-982 (Figure 5A,B). Akt and phosphorylated Akt expression did not differ between the hrTNFα and vehicle groups (Figure 5A,D,E). However, Akt expression was reduced in the presence of 125 (*p* = 0.006) ng/mL MED compared to the hrTNFα groups in SW-982 (Figure 5A,D). The level of p-Akt was also reduced in the presence of 62.5 (*p* = 0.037), 125 (*p* = 0.045), and 250 (*p* = 0.035) ng/mL MED compared to the hrTNFα groups (Figure 5A,E).

Consistent with these results using SW982, expression of p65 did not differ between the hrTNFα and vehicle groups (Figure 6A,B), and phosphorylation of p65 was increased in the presence of hrTNFα in HSF (*p* = 0.004, Figure 6A,C). Phosphorylation of p65 was suppressed in the presence of 250 ng/mL MED (*p* < 0.001, Figure 6A,B). Akt expression was reduced in the presence of 250 ng/mL MED compared to the vehicle (*p* = 0.037) and hrTNFα alone (*p* = 0.003). However, contrary to the results in SW982, p-Akt was increased in presence of 62.5 (*p* = 0.040), 125 (*p* = 0.002), and 250 (*p* = 0.036) ng/mL MED.

## 3. Discussion

In this study, we found that MED treatment led to a reduction in the upregulation of 1167 of 1478 genes which were significantly upregulated in a synovial cell line, SW-982, as well as primary human synovial fibroblasts (HSF) stimulated with TNFα. Pathway analysis revealed DEGs were enriched in the TNF and NFκB signaling pathways, further supporting the involvement of MED in modulating inflammatory responses. Further, MED inhibited the expression of *IL6* and *IL8* at both the mRNA and protein levels in both SW982 cells and HSF, and reduced p65 phosphorylation in both cell types, indicating its inhibitory effect on NFκB activation. These findings suggest that MED suppresses TNFα-mediated inflammatory cytokine production by targeting p65 phosphorylation and highlight the potential therapeutic value of MED in managing inflammatory conditions in OA.

Many studies have reported that increased proinflammatory cytokines are associated with OA severity and symptoms [24,25,26,27,28,29]. IL1β concentration in SF correlates with radiographic OA grade [28]. Increased *IL6* concentration in synovial fluid was observed in end-stage OA and its concentration correlated with pain in OA patients [24,26]. Additionally, serum *IL6* concentration is associated with cartilage loss [29]. Previous research has established a connection between the level of *IL8* in synovial fluid and severity of OA, while no such association has been found with *IL8* levels in serum [25]. In contrast, Ruan et al. demonstrated a certain correlation between serum IL8 levels and the clinical and radiological assessment of OA severity [25,27]. Here, we observed that MED effectively suppressed the production of proinflammatory cytokines IL6 and IL8 in human synovial cells stimulated with TNFα. Further investigations utilizing animal models of OA could shed light on the therapeutic potential of this agent for the treatment of OA.

Under normal conditions, NFκB is inactive and sequestered in the cytosol through its interaction with the inhibitory protein IκB [30,31]. Phosphorylation of p65 at specific sites is crucial for terminating the transcriptional activity of NFκB in the cell nucleus [7]. Our findings indicate that MED inhibits p65 phosphorylation in both SW-982 and HSF. NFκb inhibition partly contributes to the reduction in inflammatory cytokines following MED treatment. In contrast, previous studies reported an interaction between Akt and the NFκB pathway. Inhibiting Akt phosphorylation has been shown to inhibit the NFκB pathway and subsequently reduce the production of proinflammatory cytokines [22,23]. Blockage of the PI3K/Akt pathway with LY294002 has been found to partially decrease the activity of TNFα-activated HSF [23]. RNA-Seq analysis revealed that *AKT3* gene expression was reduced by MED treatment in SW982 cells. Consistent with RNA-Seq analysis, Akt protein expression levels were reduced by MED treatment in SW982 and its reduction was observed in HSF. In contrast, MED suppressed Akt phosphorylation in SW982, whereas MED increased this phosphorylation in HSF. Our results suggest that MED modulates not only Akt phosphorylation but also *AKT* transcriptional activity. However, a differential effect on Akt phosphorylation was observed between SW982 and HSF. Therefore, the relationship between the modulation of Akt and the anti-inflammatory effect by MED remains unclear. Additional experiments are needed to identify the key pathway modulated by MED.

In conclusion, MED demonstrated the ability to inhibit the production of proinflammatory cytokines in response to TNFα stimulation. MED also inhibited NFκB phosphorylation. Further research using articular chondrocytes and animal models of osteoarthritis (OA) is needed to explore the therapeutic potential of MED in the treatment of OA.

## 4. Materials and Methods

### 4.1. Purification of Medermycin

Medermycin from *Streptomyces tanashiensis* 144 was obtained by a previously described method [18]. The compound was initially reported as luteomycin [15], but we identified it as medermycin [32] by ^1^H and ^13^C NMR analyses (Appendix A).

### 4.2. Antibodies

Anti-Akt (cat no. 4691), phospho-Akt (cat no. 4060), p65 (cat no. 6956), and phospho-p65 antibody (cat no. 76778) were purchased from Cell Signaling Technology (Beverly, MA, USA). Anti-GAPDH (cat no. 014-25524), HRP-linked anti-mouse IgG (cat no. 115-035-146), and HRP-conjugated anti-mouse IgG (cat no. 115-035-003) antibody were purchased from FUJIFILM Wako Pure Chemical Co. (Osaka, Japan).

### 4.3. Cell Culture

SW-982 cells were obtained from the American Type Culture Collection (Rockville, MD, USA). Primary human synovial fibroblasts (HSFs) derived from OA patients were acquired from Sigma Aldrich (Sigma-Aldrich, St. Louis, MO, USA). Both cell types were cultured in Dulbecco’s Modified Eagle Medium (DMEM) supplemented with 10% fetal bovine serum at a temperature of 37 °C.

### 4.4. RNA Sequencing

The impact of MED on TNFα-stimulated SW-982 cells was analyzed using an RNA sequencing assay. SW-982 cells were plated at a concentration of 2 × 10^5^ cells/well in a 6-well plate and incubated for 72 h. The cells were then treated with either DMEM (vehicle) or 10 ng/mL human recombinant TNFα (hrTNFα) along with 250 ng/mL of MED for a duration of 6 h. The concentration of hrTNFα and time point in each experiment was determined based on our previous study [14]. The lactate dehydrogenase assay suggested that MED concentrations above 500 ng/mL have cytotoxicity. Therefore, a concentration of up to 250 ng/mL MED was selected for the experiments. Total RNA was extracted using Trizol (Invitrogen, Carlsbad, CA, USA) as the lysis buffer, followed by purification using the Direct-zol MicroPrep kit from Zymo Research (Orange, CA, USA). The quantity of RNA was measured using a spectrophotometer (Denovix, Wilmington, DE, USA), and its quality was assessed using an Agilent 2100 Bio Analyzer (Agilent, Santa Clara, CA, USA) through microcapillary electrophoresis. The extracted RNA was then used for RNA sequencing analysis. The RNA sequencing procedure was conducted using an MGI DNB-SEQ-G400 sequencer from BGI (Shenzhen, China). Two replicate samples were obtained from the vehicle, hrTNFα, and hrTNFα + MED groups for the purpose of RNA sequencing analysis.

### 4.5. Real Time RT-PCR (rtRT-CR)

We used rtRT-PCR to measure levels of *IL1B*, *IL6*, and *IL8* expression in the cells to validate the RNA sequencing results and investigate the effects of MED concentrations on expression levels. SW-982 and HSF were treated with either vehicle or 10 ng/mL hrTNFα in the presence of various concentrations of MED (0, 31.3, 62.5, 125, and 250 ng/mL) for 6 h (*n* = 6). First-strand cDNA was synthesized from purified total RNA after the extraction procedure was carried out as described above using the SuperScript^®^ III First-Strand Synthesis System (Invitrogen, Carlsbad, CA, USA). Quantitative PCR was carried out using the SYBR green method. Primers used in this study have been described previously [14]. A delta-delta CT method was used to determine gene expression (Gene/GAPDH) in the vehicle, and relative expression level was calculated when the average level of gene expression (Gene/GAPDH) was 1.

### 4.6. Enzyme-Linked Immunosorbent Assay (ELISA)

SW-982 cells and HSF were plated at a density of 1 × 10^4^ cells/well in 96-well plates. The SW-982 cells were then treated with 10 ng/mL hrTNFα in the presence of different concentrations of MED (0, 31.25, 62.5, 125, and 250 ng/mL) for 24 h (*n* = 6). The culture supernatants were collected, and the concentrations of IL1β, IL6, and IL8 in the supernatants were measured using commercially available ELISA kits (BioLegend, San Diego, CA, USA).

### 4.7. Western Blotting

SW-982 cells and HSF were then treated with 10 ng/mL hrTNFα along with varying concentrations of MED (0, 31.25, 62.5, 125, and 250 ng/mL) for 6 h (*n* = 4). After treatment, total protein was extracted from the samples using sodium dodecyl sulfate (SDS) sample buffer. Cell lysate (3 μg) was separated on SDS-PAGE and then transferred onto a polyvinylidene difluoride membrane. Following blocking with 10% skim milk, the membrane was reacted with anti p65, p-p65, Akt, p-Akt, or GAPDH antibody. Following the primary antibody reaction, the membrane reacted with the HRP-linked secondary antibody. The protein bands on the membrane were visualized using enhanced chemiluminescence (Chemi-Lumi One L; Nacalai Tesque, Kyoto, Japan). The luminescent images were captured using a CCD image and then subjected to luminescent image analysis. Relative expression levels of the target proteins were normalized to the expression of GAPDH, a commonly used housekeeping protein. This analysis was performed using ImageJ software.

### 4.8. Statistical Analysis

The normality of the data distribution was assessed using the Shapiro-Wilk test. Subsequently, differences among the vehicle-, hrTNFα-, and hrTNFα-, and MED-treated cells were compared with either the Kruskal–Wallis test (a non-parametric test) or multiple comparisons test with Bonferroni correction. These statistical analyses were performed using SPSS software version 25.0 from IBM (New York, NY, USA). All statistical tests were two-sided, and a *p*-value less than 0.05 was considered to indicate statistical significance.

## Figures and Tables

**Figure 1 ijms-24-13871-f001:**
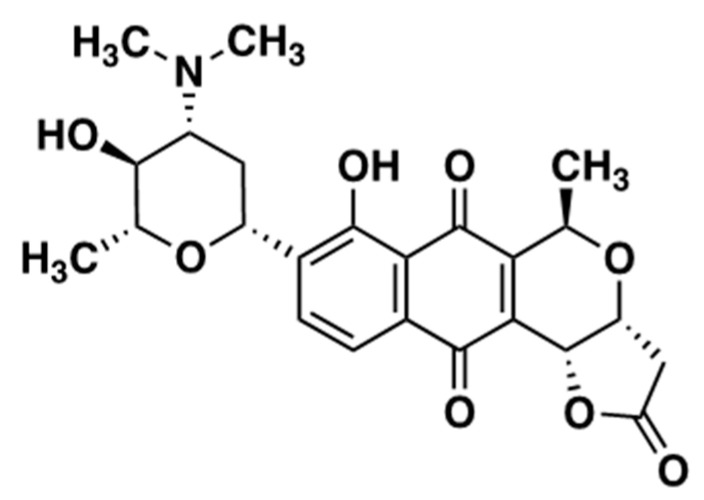
Structure of Medermycin.

**Figure 2 ijms-24-13871-f002:**
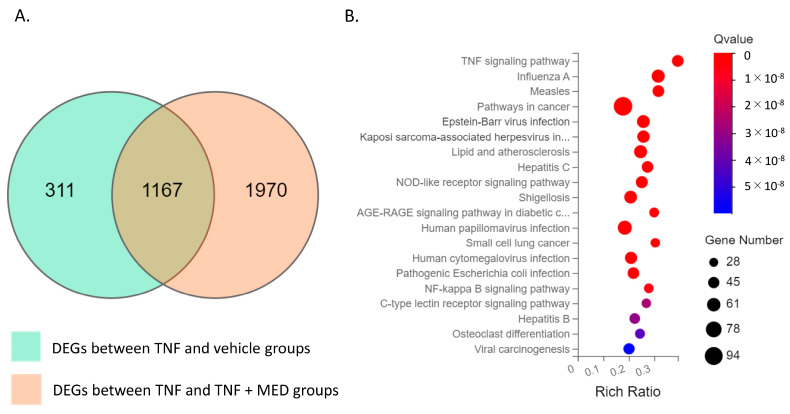
RNA sequencing analyses of vehicle, hrTNFα, and hrTNFα + medermycin (MED) groups. (**A**) Venn diagram analysis of DEGs (differentially expressed genes) showing the overlapping region between the hrTNFα/vehicle dataset and the hrTNFα + Medermycin (MED)/hrTNFα dataset. This indicates common genes that are differentially expressed in both conditions. (**B**) Bubble diagram representing the enrichment analysis of 20 KEGG pathways associated with the DEGs. Each bubble represents a specific pathway, and the size of the bubble corresponds to the number of genes involved in that pathway. Two independent experiments were carried out for each analysis.

**Figure 3 ijms-24-13871-f003:**
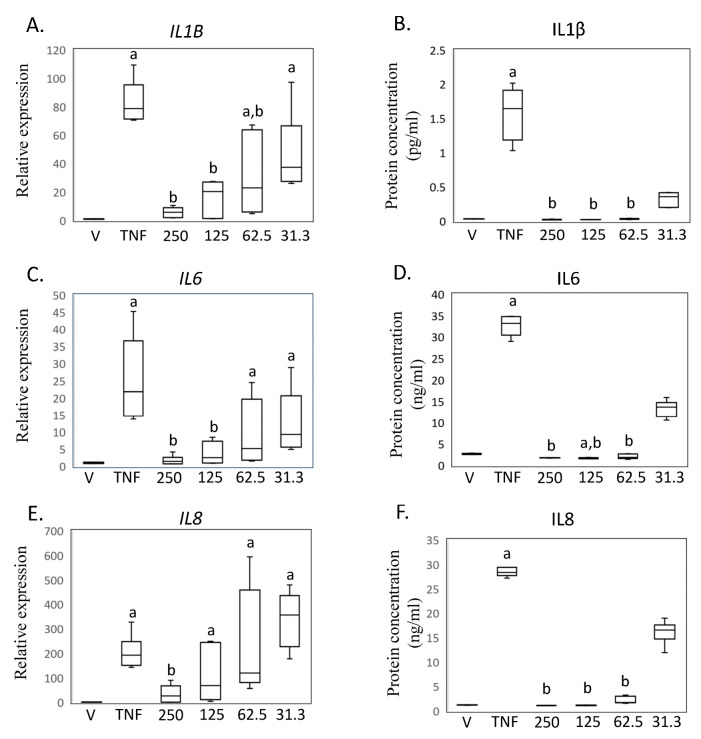
Impact of medermycin on proinflammatory cytokine expression and production in SW-982. *IL1B* mRNA by rtRT-PCR (**A**) and IL1β protein concentration by ELISA (**B**). *IL6* mRNA by rtRT-PCR (**C**) and IL6 protein concentration by ELISA (**D**). *IL8* mRNA by rtRT-PCR (**E**) and IL8 protein concentration by ELISA (**F**). SW-982 was treated with DMEM (vehicle), hrTNFα, or hrTNFα + medermycin (31.3, 62.5, 125, 250 ng/mL). ^a^ *p* < 0.05 compared to vehicle, ^b^ *p* < 0.05 compared to hrTNFα. Three independent experiments were carried out for each analysis.

**Figure 4 ijms-24-13871-f004:**
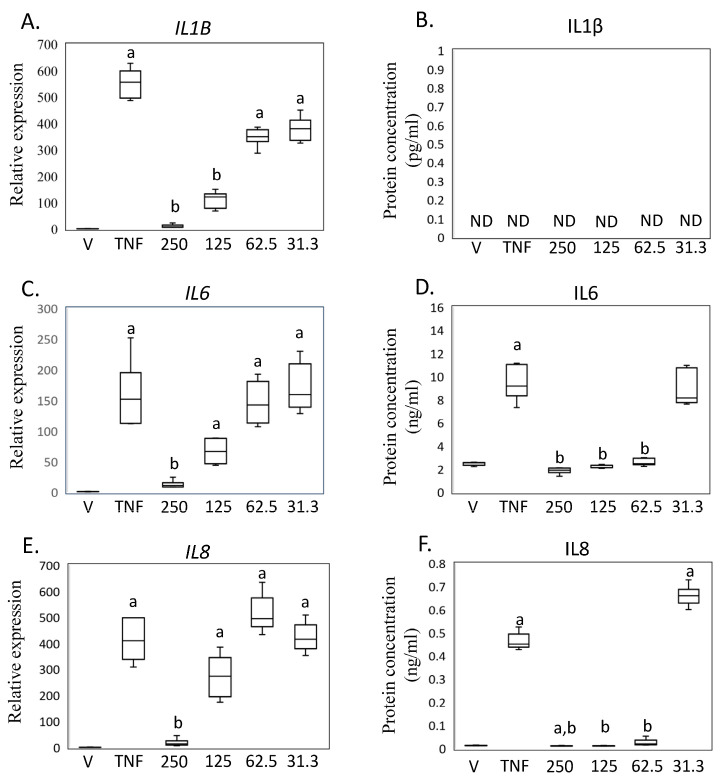
Impact of medermycin on proinflammatory cytokine expression and production in primary human synovial fibroblasts (HSFs). *IL1B* mRNA by rtRT-PCR (**A**) and IL1β protein concentration by ELISA (**B**). *IL6* mRNA by rtRT-PCR (**C**) and IL6 protein concentration by ELISA (**D**). *IL8* mRNA by rtRT-PCR (**E**) and IL8 protein concentration by ELISA (**F**). HSF treated with DMEM (vehicle), hrTNFα, or hrTNFα + medermycin. ^a^ *p* < 0.05 compared to vehicle, ^b^ *p* < 0.05 compared to hrTNFα. Three independent experiments were carried out for each analysis.

**Figure 5 ijms-24-13871-f005:**
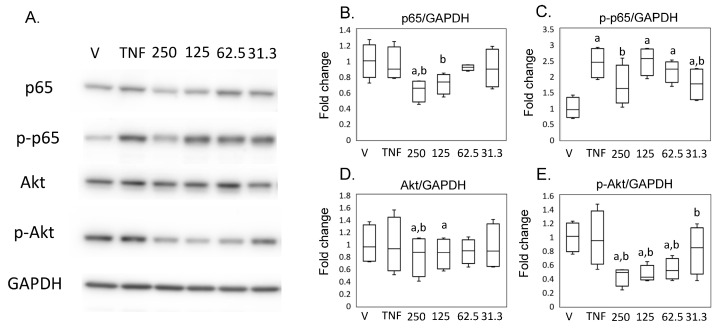
Impact of medermycin on the NFκB and Akt pathway in SW-982. Western blotting was performed to analyze the protein expression levels of p65, phosphorylated p65 (p-p65), Akt, phosphorylated Akt (p-Akt), and GAPDH (**A**) (*n* = 4). Densitometry analysis of the Western blot bands was conducted for p65 (**B**), p-p65 (**C**), Akt (**D**), and p-Akt (**E**), normalized to expression of GAPDH. SW-982 was stimulated with different treatments: DMEM (vehicle), hrTNFα alone, or hrTNFα in combination with medermycin (at concentrations of 31.3, 62.5, 125, and 250 ng/mL). ^a^ *p* < 0.05 compared with vehicle, ^b^ *p* < 0.05 compared to hrTNFα groups. Three independent experiments were carried out for each analysis.

**Figure 6 ijms-24-13871-f006:**
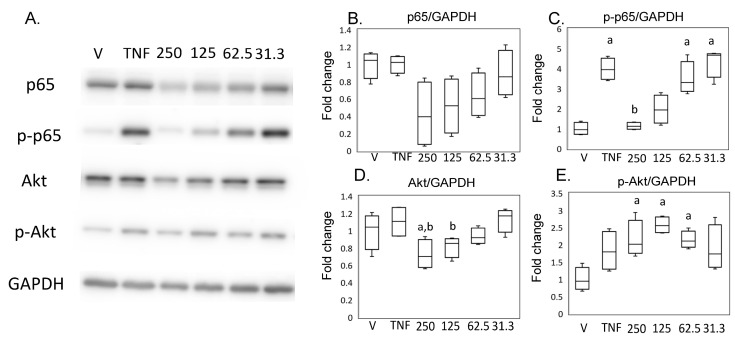
Impact of medermycin on the NFκB and Akt pathway in human primary synovial fibroblasts (HSFs). Western blotting was performed to analyze the protein expression levels of p65, phosphorylated p65 (p-p65), Akt, phosphorylated Akt (p-Akt), and GAPDH (**A**) (*n* = 4). Densitometry analysis of the western blot protein bands for p65 (**B**), p-p65 (**C**), Akt (**D**), and p-Akt (**E**) was conducted, with normalization to the expression of GAPDH. HSF were treated with different treatments: DMEM (vehicle), hrTNFα alone, or hrTNFα in combination with medermycin (at concentrations of 31.3, 62.5, 125, and 250 ng/mL). ^a^ *p* < 0.05 compared with vehicle, ^b^ *p* < 0.05 compared to hrTNFα groups. Three independent experiments were carried out for each analysis.

## Data Availability

The data presented in this study are available on request from the corresponding author. The raw RNA sequencing data has been deposited in the DNA Data Bank of Japan (DDBJ) and can be accessed with the assigned accession number DRA016359.

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
