# Peer review of "Medermycin Inhibits TNFα-Promoted Inflammatory Reaction in Human Synovial Fibroblasts"

_ijms, 2023, doi:10.3390/ijms241813871_

Round 1
Reviewer 1 Report
In the manuscript: “Medermycin inhibits TNFα-promoted inflammatory reaction in human synovial fibroblasts”, the authors discussed about MED on TNFα-induced inflammatory reactions in a synovial cell line.
Overall, this manuscript results very interesting, the authors clearly explain the rational of the study and discussed the topic point by point.
However, we would like to invite the authors to clarify some minor points:
1. Please check the check punctuation and spaces;
2. In the first paragraph of introduction the authors generally described OA disease and the inflammatory process. However, it should be useful also briefly introduce the principal therapeutic approaches (with medical devices and food supplements). In this respect, the following reference should be useful; Stellavato A, Restaino OF, Vassallo V, Cassese E, Finamore R, Ruosi C, Schiraldi C. Chondroitin Sulfate in USA Dietary Supplements in Comparison to Pharma Grade Products: Analytical Fingerprint and Potential Anti-Inflammatory Effect on Human Osteoartritic Chondrocytes and Synoviocytes. Pharmaceutics. 2021 May 17;13(5):737. doi: 10.3390/pharmaceutics13050737. PMID: 34067775; PMCID: PMC8156081;
3. Within the introduction, the authors explained the role of synovial inflammation during OA process, maybe the relation between synoviocytes and chondrocytes has to be briefly introduced;
4. Page 2, lines 55-57; the authors said: “Accumulated evidence suggests that some antibiotic agents and their derivatives have anti-inflammatory properties [11-14]. For example, the nematocidal antibiotic jietacin and its derivative exert anti-inflammatory properties via the suppression of NFκB in synovial cells [11]”. Also effects on chondrocytes are proved? Anti-inflammatory effect on synovial cells in enough to solve the OA pro-inflammatory process? There are related references? If yes, please add them;
5. Within materials and methods section the authors said: “Primary human synovial fibroblasts (HSFs) were acquired from Sigma Aldrich (Sigma-Aldrich, St. Louis, MO, USA)”. Why did not you isolated fresh cells from synovia or synovial fluid? The cells that you used are considered “healthy or not”? please, specify in the text;
6. How many western blotting were performed? Please specify;
7. In conclusion, it is possible the MED in effective in OA treatments? What is the right dose?
minor spelling mistakes are present
Author Response
Reviewer 1
Comments and Suggestions for Authors
In the manuscript: “Medermycin inhibits TNFα-promoted inflammatory reaction in human synovial fibroblasts”, the authors discussed about MED on TNFα-induced inflammatory reactions in a synovial cell line.
Overall, this manuscript results very interesting, the authors clearly explain the rational of the study and discussed the topic point by point.
However, we would like to invite the authors to clarify some minor points:
Response: Thank you for reviewing our manuscript and for these kind comments.
- Please check the check punctuation and spaces;
Response: We have carefully checked and corrected these.
- In the first paragraph of introduction the authors generally described OA disease and the inflammatory process. However, it should be useful also briefly introduce the principal therapeutic approaches (with medical devices and food supplements). In this respect, the following reference should be useful; Stellavato A, Restaino OF, Vassallo V, Cassese E, Finamore R, Ruosi C, Schiraldi C. Chondroitin Sulfate in USA Dietary Supplements in Comparison to Pharma Grade Products: Analytical Fingerprint and Potential Anti-Inflammatory Effect on Human Osteoartritic Chondrocytes and Synoviocytes. Pharmaceutics. 2021 May 17;13(5):737. doi: 10.3390/pharmaceutics13050737. PMID: 34067775; PMCID: PMC8156081;
Response: We have added sentences regarding the principal therapeutic approaches, including the citation above. (line 40 - 44)
- Within the introduction, the authors explained the role of synovial inflammation during OA process, maybe the relation between synoviocytes and chondrocytes has to be briefly introduced;
Response: We have added a sentence regarding the relation between synoviocytes and chondrocytes. (line 49 - 51)
- Page 2, lines 55-57; the authors said: “Accumulated evidence suggests that some antibiotic agents and their derivatives have anti-inflammatory properties [11-14]. For example, the nematocidal antibiotic jietacin and its derivative exert anti-inflammatory properties via the suppression of NFκB in synovial cells [11]”. Also effects on chondrocytes are proved? Anti-inflammatory effect on synovial cells in enough to solve the OA pro-inflammatory process? There are related references. If yes, please add them;
Response: Synovial inflammation is one of the therapeutic targets for OA. However, the effect on chondrocytes is also important for OA treatment. In addition to animal experiments, further investigation using chondrocytes is needed. We have added this point in the Discussion section. (line 33, 209)
- Within materials and methods section the authors said: “Primary human synovial fibroblasts (HSFs) were acquired from Sigma Aldrich (Sigma-Aldrich, St. Louis, MO, USA)”. Why did not you isolated fresh cells from synovia or synovial fluid? The cells that you used are considered “healthy or not”? please, specify in the text;
Response: Cell culture was performed at ÅŒmura Satoshi Memorial Institute, Kitasato University, Tokyo, Japan. This institute is not able to obtain fresh cells in its own institute. Therefore, HSFs were obtained from Sigma-Aldrich. We used cells originating from OA patients. We have added a sentence stating that the cells originated from OA patients. (line 224 - 225)
- How many western blotting were performed? Please specify;
Response: We performed western blotting using four samples. We have added this point.
- In conclusion, it is possible the MED in effective in OA treatments? What is the right dose?
Response: Further investigations utilizing animal models of OA may provide valuable insights into the therapeutic potential of MED for this disease. However, the question of dose remains unclear.
Reviewer 2 Report
ijms-2563554
Medermycin inhibits TNFα-promoted inflammatory reaction in human synovial fibroblasts
This manuscript focuses on the effect of the Streptomyces-derived antibiotic medermycin (MED) on synovial inflammation in osteoarthritis (OA) using TNF-stimulated fibroblasts as a model (i.e., primary human synovial fibroblasts (HSF) and the synovial cell line SW-982). Based on RNA sequencing data, the authors report that 1478 genes were upregulated in TNF-stimulated SW-982 cells, while MED significantly reduces the expression of 1167 of these genes (including IL1B, IL6, and IL8). KEGG pathway analyses revealed that these genes were associated with TNF and NF-κB signaling. Moreover, different concentrations of MED proved to reduce mRNA and protein levels of IL1B, IL6, and IL8, level and phosphorylation of p-65, and levels of Akt in TNF-treated HSF and SW-982 cells. In the latter, MED also reduces Akt phosphorylation, while in HSF, a MED-dependent increase in Akt phosphorylation was observed. The authors conclude that MED possesses promising anti-inflammatory properties and may represent a potential drug to manage the inflammatory processes in OA.
The study is carried out properly and technically/statistically sound. The results shown are comprehensible and conclusive. Text and figures are straightforward and clear. Moreover, the use of primary HSF (in addition to the SW-982 cell line) is an advantage of the study. However, there are some critical points requiring the authors’ consideration.
Major comments:
In my opinion, the study has two major problems.
1. The experiments have been performed using the synovial cell line SW-982 and normal primary HSF only. Though the cells have been treated with TNF to generate an activated, inflamed phenotype, it is not clear whether this adequately reflects the inflammatory state present in osteoarthritis. Therefore, the key experiments have to be reproduced using primary HSF from patients suffering from OA (or a comparable inflammatory disease involving HSF).
2. Since the results concerning the involvement of Akt obtained in SW-982 and primary HSF differ, it is not clear by which pathway the anti-inflammatory effect of MED and the associated downregulation of p65 phosphorylation is mediated. Additional experiments have to be performed to provide further mechanistic insight and to identify (the) key pathway(s) modulated by MED.
Minor comments:
3. A few missing/redundant spaces in the text should be corrected.
4. Results: For a better overview, the relevant basic experimental conditions should be included in the text (e.g., TNF concentration, time points analysed, ...).
5. Please include information on the fold changes of expression and phosphorylation levels following TNF and MED treatment, respectively.
6. The number of independent experiments should be indicated in the Figure Legends.
7. According to the Figures, p-p65 and p-Akt levels are also reduced in the presence of 31,3 ng/ml MED (Fig. 4C and E) and Akt levels appear to be reduced in the presence of 125 ng/ml MED (Fig. 5D) when compared to TNF. This should be mentioned/discussed in the text.
8. Figure 5C: Is it correct that 125 ng/ml MED do not significantly reduce p65 phosphorylation in comparison to the TNF-treated sample?
9. Materials and Methods: A Figure showing the molecular structure of MED should be included.
10. Line 217: Please indicate the TNF concentration used. Did the authors use the same concentration for all experiments, or did the concentration differ among different experimental approaches?
11. Line 217: Why was a concentration of/up to 250 ng/ml MED selected for the experiments? Did the authors perform dose response experiments before? If so: Which read-out was used?
Author Response
Reviewer 2
Comments and Suggestions for Authors
ijms-2563554
Medermycin inhibits TNFα-promoted inflammatory reaction in human synovial fibroblasts
This manuscript focuses on the effect of the Streptomyces-derived antibiotic medermycin (MED) on synovial inflammation in osteoarthritis (OA) using TNF-stimulated fibroblasts as a model (i.e., primary human synovial fibroblasts (HSF) and the synovial cell line SW-982). Based on RNA sequencing data, the authors report that 1478 genes were upregulated in TNF-stimulated SW-982 cells, while MED significantly reduces the expression of 1167 of these genes (including IL1B, IL6, and IL8). KEGG pathway analyses revealed that these genes were associated with TNF and NF-κB signaling. Moreover, different concentrations of MED proved to reduce mRNA and protein levels of IL1B, IL6, and IL8, level and phosphorylation of p-65, and levels of Akt in TNF-treated HSF and SW-982 cells. In the latter, MED also reduces Akt phosphorylation, while in HSF, a MED-dependent increase in Akt phosphorylation was observed. The authors conclude that MED possesses promising anti-inflammatory properties and may represent a potential drug to manage the inflammatory processes in OA.
The study is carried out properly and technically/statistically sound. The results shown are comprehensible and conclusive. Text and figures are straightforward and clear. Moreover, the use of primary HSF (in addition to the SW-982 cell line) is an advantage of the study. However, there are some critical points requiring the authors’ consideration.
Major comments:
In my opinion, the study has two major problems.
Response: Thank you for reviewing our manuscript and for your valuable comments.
- The experiments have been performed using the synovial cell line SW-982 and normal primary HSF only. Though the cells have been treated with TNF to generate an activated, inflamed phenotype, it is not clear whether this adequately reflects the inflammatory state present in osteoarthritis. Therefore, the key experiments have to be reproduced using primary HSF from patients suffering from OA (or a comparable inflammatory disease involving HSF).
Response: We used human primary HSF derived from OA patients. We have added this information in the Materials and methods. (line 224 - 225)
- Since the results concerning the involvement of Akt obtained in SW-982 and primary HSF differ, it is not clear by which pathway the anti-inflammatory effect of MED and the associated downregulation of p65 phosphorylation is mediated. Additional experiments have to be performed to provide further mechanistic insight and to identify (the) key pathway(s) modulated by MED.
Response: We agree with reviewer’s comments. To avoid overreaching, we have revised the sentences in the Discussion and Conclusion sections. (line 191 - 206)
Minor comments:
- A few missing/redundant spaces in the text should be corrected.
Response: We have carefully checked and correct these.
- Results: For a better overview, the relevant basic experimental conditions should be included in the text (e.g., TNF concentration, time points analysed, ...).
Response: TNF concentration and time points were determined based on previous our study.
We have added this point in the Materials and methods section. (line 233 - 236)
- Please include information on the fold changes of expression and phosphorylation levels following TNF and MED treatment, respectively.
Response: We have revised the Y-axis to include information on fold changes in expression in new Figures 5 and 6.
- The number of independent experiments should be indicated in the Figure Legends.
Response: Two (RNA-seq) or three (PCR, ELISA, western blot) independent experiments were carried out for each analysis. We have added this point in the Figure legends. (line 88, 101, 127, 147-148, 163)
- According to the Figures, p-p65 and p-Akt levels are also reduced in the presence of 31,3 ng/ml MED (Fig. 4C and E) and Akt levels appear to be reduced in the presence of 125 ng/ml MED (Fig. 5D) when compared to TNF. This should be mentioned/discussed in the text.
Response: Thank you for insightful comment. We have this point in the Discussion section. (line 197 - 206)
- Figure 5C: Is it correct that 125 ng/ml MED do not significantly reduce p65 phosphorylation in comparison to the TNF-treated sample.
Response: This is correct. The P value is 0.089.
- Materials and Methods: A Figure showing the molecular structure of MED should be included.
We have added the molecular structure of MED in a new Figure 1.
- Line 217: Please indicate the TNF concentration used. Did the authors use the same concentration for all experiments, or did the concentration differ among different experimental approaches?
Response: We used 10 ng/ml TNF for all experiments. We have added this concentration in the figure legends and Methods section.
- Line 217: Why was a concentration of/up to 250 ng/ml MED selected for the experiments? Did the authors perform dose response experiments before? If so: Which read-out was used?
Response: Lactate dehydrogenase assay suggested that an MED concentration above 500 ng/ml has cytotoxicity. Therefore, a concentration of up to 250 ng/ml MED was selected for the experiments. We have added this point in the Materials and methods section. (line 234 - 236)
Round 2
Reviewer 2 Report
ijms-2563554
The manuscript provides a revised version of the study “Medermycin inhibits TNFα-promoted inflammatory reaction in human synovial fibroblasts”. The manuscript has been improved and my comments have been adequately addressed.